# Risk for Depressive Symptoms among Hospitalized Women in High-Risk Pregnancy Units during the COVID-19 Pandemic

**DOI:** 10.3390/jcm9082449

**Published:** 2020-07-31

**Authors:** Shanny Sade, Eyal Sheiner, Tamar Wainstock, Narkis Hermon, Shimrit Yaniv Salem, Tamar Kosef, Talya Lanxner Battat, Sharon Oron, Gali Pariente

**Affiliations:** 1Department of Obstetrics and Gynecology, Soroka University Medical Center, Ben-Gurion University of the Negev, Beer-Sheva 8457108, Israel; sheiner@bgu.ac.il (E.S.); nicky.narkis@gmail.com (N.H.); Yanivshi@bgu.ac.il (S.Y.S.); talyalan@gmail.com (T.L.B.); sharonmelamed1@gmail.com (S.O.); galipa@bgu.ac.il (G.P.); 2Department of Public Health, Faculty of Health Sciences, Ben-Gurion University of the Negev, Beer-Sheva 8457108, Israel; wainstoc@bgu.ac.il; 3Department of psychiatry, Soroka University Medical Center, Ben-Gurion University of the Negev, Beer-Sheva 8457108, Israel; tamar.kosef@gmail.com

**Keywords:** COVID-19, depression, EPDS, pandemic, pregnancy

## Abstract

Objective: Higher rates of mental disorders, specifically depression, were found among affected people in previous epidemiological studies taken after disasters. The aim of the current study was to assess risk for depression among pregnant women hospitalized during the “coronavirus disease 2019” (COVID-19) pandemic, as compared to women hospitalized before the COVID-19 pandemic. Study design: A cross-sectional study was performed among women hospitalized in the high-risk pregnancy units of the Soroka University Medical Center (SUMC). All participating women completed the Edinburgh Postnatal Depression Scale (EPDS), and the results were compared between women hospitalized during the COVID-19 strict isolation period (19 March 2020 and 26 May 2020) and women hospitalized before the COVID-19 pandemic. Multivariable logistic regression models were constructed to control for potential confounders. Results: Women hospitalized during the COVID-19 strict isolation period (*n* = 84) had a comparable risk of having a high (>10) EPDS score as compared to women hospitalized before the COVID-19 pandemic (*n* = 279; 25.0% vs. 29.0%, *p* = 0.498). These results remained similar in the multivariable logistic regression model, while controlling for maternal age, ethnicity and known mood disorder (adjusted odds ratio (OR) 1.0, 95% CI 0.52–1.93, *p* = 0.985). Conclusion: Women hospitalized at the high-risk pregnancy unit during the COVID-19 strict isolation period were not at increased risk for depression, as compared to women hospitalized before the COVID-19 pandemic.

## 1. Introduction

The “coronavirus disease” (COVID-19) caused by Severe Acute Respiratory Syndrome Coronavirus 2 (SARS-COV-2), was first isolated in January 2020, after a series of respiratory infections of unknown etiology were detected in China [1,2].

SARS-COV-2 was first diagnosed in Israel on 21 February 2020, among Israelis returning from abroad or from those who came in contact with infected tourists. According to the World Health Organization (WHO) report from June, 823,813 cases have been diagnosed in Israel since then [3]. At the onset of the study, 19 March, there were 529 cases of COVID-19 in Israel, with daily change of 23.89% from the previous day. In comparison, in Italy at the same day there were 41,035 cases, with a daily rise of 14.9%, and in Spain, 3431 cases, with a daily rise of 35.19% [3]. The Israeli Ministry of Health recommended soon after that all citizens returning from Eastern countries, and later on, from all other countries, stay in quarantine for two weeks following their return, in order to minimize the contagion. Not long after, on 12 March, preschools, schools and higher education closed, public transportation was halted and social isolation was implemented. Since March 2020, using satellite information and cellular phone location, the Israeli Ministry of Health sent automated text messages to individuals identified as being close to positively diagnosed citizens, informing them that they should stay in quarantine and contact a health care provider in the case that any symptom develops.

Previous epidemiological studies were taken in order to assess the effect of a pandemic on the mental health of affected people. During the 2003 (Severe Acute Respiratory syndrome (SARS) pandemic, studies have demonstrated depression, anxiety, panic attack, psychotic symptoms, delirium and even suicidal ideations among the pandemic survivors [4]. The survivors also experienced psychological effects of physical symptoms, such as anxiety and insomnia regarding fever and dysphoria due to nausea [5].

Likewise, the sudden outbreak of COVID-19, the unpredictability of the situation, quarantine for indefinite periods, myths and misinformation about the epidemic, the unavailability of vaccine and the overflow of information on social media have affected the public social health [4,6], and even provoked extreme behavior like suicidal ideations [7].

The onset of depression is reported to peak during childbearing years and is approximately twice more common in women than in men [8]. Pregnancy and postpartum period are vulnerable times for onset or relapse of mental illness, with depression and anxiety being the most common psychiatric disorders [9]. Studies have demonstrated serious concerns of depressive symptoms during pregnancy, in terms of maternal morbidity and adverse neonatal outcomes [10]. Pregnant women hospitalized in a high-risk pregnancy unit are described to have fair risk for depression, varying from 27–44% [11,12,13].

With little existing published data, focusing on depression among pregnant women hospitalized during the COVID-19 pandemic, and in light of its potential adverse effect on both the mother and the infant, we aimed to assess the incidence of depression among women hospitalized in the high-risk units during the COVID-19 strict isolation period, as compared with the incidence of depression among women hospitalized in the high-risk units before the COVID-19 pandemic.

## 2. Materials and Methods

### 2.1. Population and Setting

The study recruited women hospitalized in high-risk units in the Soroka University Medical Center (SUMC). Recruitment and data collection were done during the COVID-19 pandemic strict isolation period, between 19 March 2020 and 26 May 2020. Women who were hospitalized in the high-risk pregnancy units before the COVID-19 pandemic (between November 2016 and April 2017) served as the comparison group. The comparison group was composed of women who were recruited for previous study performed in the same high-risk pregnancy units [11]. High risk pregnancy was defined according to the Israeli Ministry of Health statement regarding criteria for high-risk pregnancy. These include maternal chronic illness (such as cardiac disease, pulmonary disease, hypertension, diabetes), obstetric and gynecological history (such as recurrent pregnancy loss, congenital abnormalities, intrauterine fetal death), and other conditions, such as multiple gestation, suspected intrauterine growth retardation, suspected macrosomia and more [14]. SUMC, located in southern Israel, is the largest country birth center, with more than 17, 000 births a year. The study was approved by the SUMC IRB Committee (IRB approval # 0079-20-SOR).

### 2.2. Study Design

A cross-sectional study was performed; women hospitalized in the high-risk pregnancy units during the COVID-19 pandemic were compared to historic unexposed group hospitalized at the same units, before the COVID-19 pandemic. Each woman participated at a single time-point. Every day, during the time frame of the study, the research team handed out self-reported questionnaires to all women who met the inclusion criteria, in the high-risk pregnancy unit, following an oral and written explanation on the study course and purpose. All hospitalized women were approached, regardless of hospitalization length or indication. Each woman from both the exposed and the comparison group, answered questions regarding her socioeconomic state, medical background, obstetrical history, current pregnancy course, and completed the Edinburgh Postnatal Depression Scale (EPDS) questionnaire. Both the exposed group and the comparison group fulfilled the questionnaires prospectively at the hospital. This screening test, established by Cox et al. in 1987 [15], was developed for diagnosing pregnant and postpartum women who are at high risk for depression. This questionnaire is widely used, based on the American College of Obstetrics and Gynecology (ACOG) recommendations [16]. According to the Israeli Ministry of Health statement from January 2014, a score of 10 and above on the EPDS indicates a high risk for depression in a pregnant woman, and requires further assessment and treatment by a mental health specialist [17]. This cutoff score has been recommended by the developers of the scale [15], and has previously been used in several other studies, as indicating a risk for both antenatal and postpartum depression [13]. The questionnaire consists of 10 self-completed questions regarding mood in the past week. The scores in each question are summed, and a final score of <10 is defined as low risk for depression. A score of ≥10 defines a person at risk for depression [15]. Background variables assessed included maternal demographic, medical and obstetrical data. Data were collected in a cross-sectional format, at two time points. Data regarding the comparison group were collected as part of a previous prospective study that was taken in the same high-risk units [11]. The group of women which served as the “unexposed” group was available from an historic study [11], and so the sample size of this group was fixed. As soon as the COVID-19 spread to Israel, and following the strict isolation regulation, the study team began to recruit women to participate in this study (following IRB approval of the study protocol). Since the isolation has gradually released, its effect could no longer be studied, and 90 women were recruited in the relevant period. Based on a priori assumptions, the given sample size would have enabled us to detect a difference of (OR = 2.17) between the study groups in the risk for EPDS ≥ 10. When planning the study, a greater difference in EPDS ≥ 10 was expected between the groups (odds ratio > 2). This big difference was not detected between the groups, however the strict isolation period has discontinued, and therefore recruitment stopped. Since recruitment timing was critical for the studied question, a power analysis was performed based on the give sample and the detected differences.

### 2.3. Definitions

Mood disorder was defined as any psychiatric diagnoses mentioned in the computerized data of each woman.

Chronic illness included diabetes mellitus, hypertension, gastrointestinal disorders, cardiac disorders, respiratory disorders, rheumatic disorders and thyroid disorders.

Positive suicidal ideation was defined as any positive answer to question number 10 (‘The thought of harming myself has occurred to me’) in the EDPS questionnaire.

### 2.4. Statistical Analysis

Statistical analysis was performed using SPSS version 23.0, IBM SPSS statistics for Windows, version 23.0. Armonk, NY: IBM Corp. Comparison of continuous variables was performed using Student’s *t*-test and Chi-square test was used to examine differences in the distribution of categorical variables. Multivariable logistic regression models were constructed to examine the association between the independent and dependent variables, while adjusting for confounding factors based on the univariate analysis, as well as clinically important variables, including maternal age, ethnicity and known mood disorder.

The strategy for model building was as follows: background characteristics were compared between the study groups (exposed and unexposed women). Variables associated with the exposure (i.e., were different between the study groups) were suspected as confounding variables, and they were tested in the multivariable models, to determine whether they are also associated with the outcome variable, and are therefore possibly confounding the association between the exposure and the outcome. A suspected confounding variable included the having a history of mood disorders. Maternal age and ethnicity are both variables with clinical significance. Ethnicity in our study population represents social, cultural, educational and religious differences between the groups, which have been known to affect EPDS scores [18,19].

## 3. Results

A total of 369 women were included in the study: 90 women who were hospitalized in the high-risk pregnancy units during the COVID-19 pandemic strict isolation period, and 279 women before the COVID-19 pandemic. The response rate of the study was 93.33%; 90 women were included in the study, 6 refused to complete the questionnaire.

EPDS questionnaires with missing data on ≥1 questions were excluded from the analysis of the total EPDS score (of EPDS ≥ 10), but were included in the comparison of responses to question 10, if this was not missing. A total of 19 questionnaires were excluded.

Table 1 summarizes maternal demographic and obstetric features of both groups. Mean maternal age was comparable between the groups, as was ethnicity and gravidity. Likewise, no significant differences were noted between the groups, in terms of fertility treatments, chronic illnesses, body mass index (BMI), bad obstetric history and indication for current hospitalization, such as suspected intra-uterine growth restriction. Rates of mood disorders were lower among pregnant women hospitalized during the COVID-19 pandemic, compared to pregnant women hospitalized before the COVID-19 pandemic.

Table 2 describes the scores of the self-completed EDPS questions. No significant differences were noted between the two groups in any of the questions.

Maternal depression risk, as assessed by the total EPDS score, is presented in Table 3. Women hospitalized during the COVID-19 pandemic had a comparable risk of having depression, expressed by a high (≥10) EPDS score, compared to women hospitalized before the COVID-19 pandemic (25.0% vs. 29.0%, *p* = 0.498). Rates of positive suicidal ideations (according to question number 10 in the EPDS questionnaire) were comparable between the groups (8.6% vs. 5.0%, *p* = 0.221).

Using a multivariable logistic regression model, controlling for maternal age, ethnicity and known mood disorder, women hospitalized during the COVID-19 pandemic had a comparable risk for depression, as compared to women hospitalized before the COVID-19 pandemic (adjusted OR 1.0, 95% CI 0.52–1.93, *p* = 0.985, Table 4). Another multivariable logistic regression model, controlling for maternal age, showed that women hospitalized during the COVID-19 pandemic had a comparable risk for suicidal ideation as compared to women hospitalized before the COVID-19 pandemic (adjusted OR 1.8, 95% CI 0.71–4.85, *p* = 0.203, Table 4).

## 4. Discussion

### 4.1. Principal Findings

Our study demonstrated a comparable risk for depression among women hospitalized in high-risk pregnancy units during the COVID-19 pandemic, compared to those hospitalized before the pandemic.

### 4.2. Results

Existing literature has demonstrated linkage between disaster exposure and mental health [20]. Studies showed that people exposed to natural disasters, including hurricanes, floods and earthquakes or specific collective traumatic events such as wars, may have series threat to mental health, with those who had higher exposure having greater rates of mental disorder [20,21]. Adverse mental health outcomes include particularly post-traumatic stress disorder, depression, or anxiety, including major depressive disorder and generalized anxiety disorder. In addition, health-related problems, such as somatic complaints, sleep disturbances, and substance abuse were reported among survivors of collective disasters [20,21,22].

Women are more vulnerable to disaster-related psychopathology than men [23,24,25]. Female gender has generally been associated with reduced resilience after disaster, but greater post-traumatic growth. After the Madrid train bombing, women reported more post-traumatic growth and positive changes, but also more negative changes and associated depression and anxiety [25].

Pregnant and post-partum women may be vulnerable to post-disaster psychopathology, and their mental health is of particular concern, because of their special role in taking care of their children and families [25]. Unplanned pregnancy, being multiparous and a poor marital relationship were associated with worse pregnancy mental health [23], while having support from the partner and family were protective [26,27]. Nevertheless, in our population, marital status was comparable between the groups.

Few studies, conducted world-wide following disasters, investigated the psychiatric morbidity of pregnant and post-partum women in the disaster areas. Khatry et al. examined the influence of the 2015 Nepal earthquake on pregnant women at the time. By using the EDPS score, they assessed clinically-significant symptoms of antenatal common mental disorders, their risks and protective factors. They found that pregnant women who experienced the earthquake had higher risk for clinically significant mental disorders. Women with greater vulnerability were those who lack intimate partner relationship, had prior pregnancy losses and who lack income-generating work [26].

Chang et al. examined the influence of Taiwan 921 earthquake on mental health in a group of women from Pu-LI, a town a few kilometers from the epicenter, who were pregnant during or immediately after the major earthquake disaster. The prevalence of minor psychiatric morbidity found was high, 29.2% of all women examined, with a positive correlation to post-traumatic stress disorder. A risk factor for psychiatric morbidity was spouse causality [27].

On the contrary, overall rates of depression and posttraumatic stress disorder were not significantly higher among groups of postpartum women from southern Louisiana who were pregnant during or shortly after the Hurricane Katrina [23]. In their next study, the authors demonstrated that, not only were the pregnant and post-partum women more resilient to the consequences of the disaster, they also grew and perceived benefits after the disaster [25].

Engel et al. investigated the consequences of the destruction of the World Trade Center on 11 September 2001 on the health of pregnant women and their fetuses. They enrolled women who were pregnant and living or working within close proximity to the disaster. They found that post-traumatic stress symptomatology and moderate depression were associated with long gestational duration [24].

As can be seen in other related studies, poor marital relationship was associated with worse pregnancy mental health [23], while having support from the partner and family were protective [26,27]. Our study failed to find such as association. This may be explained by the size of the cohort, or by the fact that our study assessed risk for depression by the EDPS questionnaire and not a diagnosis of depression. Results may have been different if we would have repeated the study few of months after delivery.

While most of the previous studies investigated post- disaster depression, our study investigated risk for depression during the COVID-19 pandemic. Mounder et al. investigated mental health response among health care workers during the first 4 weeks of the SARS outbreak in Toronto, Canada. The authors found that the most prominent emotional effects were fear, loneliness, boredom, anger, and worry about the effect of quarantine and contagion; they experienced the psychological effect of physical symptoms [5].

### 4.3. Clinical Implications

Our study demonstrated comparable rates of depression among women hospitalized in the high-risk pregnancy units during and before the COVID-19 pandemic. There are several possible explanations for this association. First, as was noted previously, many people, and especially pregnant and postpartum women, are resilient after terrible events, and may experience various forms of benefits [25]. Posttraumatic benefit relates to posttraumatic growth going beyond baseline to an improved state of functioning after trauma. Such growth is described as changed priorities, having a greater appreciation of life, an increased sense of strength, self-reliance, expressiveness, compassion, and improved relationships. Second, Harville et al. determined that greater experience of the disaster was associated with less resilience [25]. Resilience is the ability to overcome difficulties and stressors, and does not mean being completely unaffected by terrible events or not having limited periods of mental health problems. It is heavily influence by how closely the disaster personally affected them. Our study population did not have any patients with a proven infection with SARS-COV-2, which may be related to the pandemic self-experience of the women and their mental consequences enrolled in our study. The same group also demonstrated that mental health problems following forces on the general population might be even lower in post-partum women, possibly because of more social support and nurturing after difficult circumstances [23]. This may also explain the comparable rates of depression between the study groups. Bonnano et al. examined the prevalence of resilience among a large group of New York area residents during the six months following the 11 September terrorist attack, demonstrated that even among the groups with the most pernicious levels of exposure and highest rates of post-traumatic stress disorder, the proportion that were resilient never dropped below one third [28]. This may explain our findings of comparable depression rates between women during and before the pandemic, with a substantial percentage of the exposed women who demonstrated resilience. Finally, the EDPS questionnaire is a rather screening then diagnostic test, hence false negative results may occur. In addition, depression may evolve months after filling the questionnaire.

### 4.4. Strength and Limitations

Strengths of the study include the use of standardized mental health instrument and a systematic recruitment of an unselected population. However, our study has limitations. First, the sample size limits the generalizability of our data. Missing data can affect the value of patient reported outcome and the precision of the estimated change in the results, which may introduce a bias. Nevertheless, in our study, the only covariate that presented missing data was BMI, with a rate of less than 5% missing data. In addition, we have no data on trends over longer periods of hospitalization. Third, EDPS is not a diagnostic tool but a screening tool, hence, we may have a false negative result, and women with actual depression may not be diagnosed and get the social and psychiatric aid that they deserve. Another limitation of our study relates to the possible seasonality effects on the studied association, since exposed and unexposed were recruited in different season, due to the COVID-19 exposure window. Another limitation of the study is the wide range of the gestational week of the women participating in the study as depressive symptoms score may vary during gestation. Nevertheless, in our study, no difference was demonstrated between the groups in mean gestational age. Information regarding the extent of medication use in these women was unavailable, including psychotropics or other unmeasured confounders that were not accounted for.

Leading theories about the emergence of depression emphasize that depression often occurs following repeated stressors or feelings of hopelessness, but the study window is limited and occurs at the onset of the pandemic. It may not be enough time for depression to develop. Nevertheless, the study’s main purpose was to assess risk for depression, rather than depression itself. As our study occurred at the onset of the pandemic, more studies should be taken in the future during later outbreaks of the COVID-19 pandemic, as leading theories about the emergence of depression emphasize that depression often occurs following repeated stressors or feelings of hopelessness.

Our study participants (both exposed and unexposed) were already hospitalized in high-risk pregnancy units. Therefore, it is possible that their own medical issues would take precedence over the stress of the pandemic, which could account for why there are no group differences. Hence, the results of the current study are not generalized to all pregnant women, as some of them may be experiencing heightened depressive symptoms during this pandemic.

### 4.5. Conclusions

In conclusion, our study found that women hospitalized in the high-risk pregnancy units during the COVID-19 pandemic have comparable risk for depression compared to the comparison group of high-risk pregnant women not hospitalized during the pandemic. More studies should be done in order to shed some more light on the association between hospitalization in the high-risk pregnancy units during the COVID-19 pandemic and depression, as well as to assess the prevalence of other mental disorders among this population during the COVID-19 pandemic.

## Figures and Tables

**Table 1 jcm-09-02449-t001:** Demographic and obstetric characteristics of the study population.

Characteristic		Hospitalization during the “Coronavirus Disease” (Coronavirus Disease 2019 (COVID-19)) Pandemic *n* = 84 (%)	Hospitalization before the COVID-19 Pandemic *n* = 279 (%)	*p* Value
Maternal age, years	<20	2.4	4.3	0.444
	20–35	79.8	82.4	
	>35	17.9	13.3	
Gestational age at hospitalization, weeks (mean + SD)		33.7 ± 5.1	34.0 ± 4.8	0.771
Ethnicity	Jewish	50.0	61.2	0.071
	Bedouins	50.0	38.8	
Marital status	Married	90.4	92.8	0.460
	Other	9.6	7.2	
Gravidity	1	28.6	26.2	0.153
	2–4	42.9	53.8	
	5≤	28.6	20.1	
Parity	1	34.5	34.8	0.116
	2–4	46.4	54.5	
	5≤	19.0	10.8	
Fertility treatments		10.7	9.5	0.738
Diabetes mellitus		11.4	9.5	0.625
Past abortions		38.6	38.8	0.972
* body mass index (BMI)	<20	0.0	1.1	0.567
	20–24.99	18.8	24.9	
	25–29.99	43.5	40.5	
	>30	37.7	33.5	
Mood disorder		10.7	20.8	0.037
Chronic illness		22.6	17.7	0.311
Premature rapture of membranes		4.8	4.7	0.969
Suspected intra-uterine growth restriction		9.5	7.5	0.554
Past intra-uterine fetal death		4.8	7.6	0.388
Past neonatal death		2.4	2.8	0.857
Past pre-eclampsia		8.5	6.2	0.494
Past congenital abnormalities		4.8	6.0	0.676

* Missing value: BMI: *n* = 25.

**Table 2 jcm-09-02449-t002:** Scores of the EDPS questions, among women hospitalized in the high-risk pregnancy units during and before the COVID-19 pandemic.

EDPS Question		Hospitalization during the COVID-19 Pandemic *n* = 84 (%)	Hospitalization before the COVID-19 Pandemic *n* = 279 (%)	*p* Value
1- I have been able to laugh and see the funny side of things	0- As much as I always could	57.5	68.5	0.068
	1- Not quite so much now	35.0	21.5	
	2- Definitely not so much now	5.0	4.3	
	3- Not at all	2.5	5.7	
2- I have looked forward with enjoyment to things	0- As much as I ever did	57.0	65.3	0.552
	1- Rather less than I used to	27.8	23.5	
	2- Definitely less than I used to	10.1	6.9	
	3- Hardly at all	5.1	4.3	
3- I have blamed myself unnecessarily when things went wrong	0- Yes, most of the time	8.5	8.0	0.533
	1- Yes, some of the time	14.6	17.4	
	2- Not very often	22.0	28.3	
	3- No, never	54.9	46.4	
4- I have been anxious or worried for no good reason	0- No, not at all	43.9	44.4	0.717
	1- Hardly ever	24.4	19.5	
	2- Yes, sometimes	23.2	28.2	
	3- Yes, very often	8.5	7.9	
5- I have felt scared or panicky for no good reason	0- Yes, quite a lot	1.2	6.5	0.250
	1- Yes, sometimes	18.5	14.0	
	2- No, not much	23.5	23.7	
	3- No, not at all	56.8	55.8	
6- Things have been getting on top of me	0- Yes, most of the time I haven’t been able to cope at all	9.0	9.3	0.964
	1- Yes, sometimes I haven’t been coping as well as usual	25.6	27.2	
	2- No, most of the time I have coped quite well	29.5	26.5	
	3- No, I have been coping as well as ever	35.9	36.9	
7- I have been so unhappy that I have had difficulty sleeping	0- Yes, most of the time	7.5	7.2	0.886
	1- Yes, sometimes	13.8	15.1	
	2- Not very often	22.5	18.6	
	3- No, not at all	56.3	59.1	
8- I have felt sad or miserable	0- Yes, most of the time	11.3	8.7	0.813
	1- Yes, quite often	10.0	11.8	
	2- Not very often	28.7	27.2	
	3- No, not at all	50.0	52.7	
9- I have been so unhappy that I have been crying	0- Yes, most of the time	3.7	6.8	0.356
	1- Yes, quite often	13.6	9.4	
	2- Only occasionally	27.2	21.9	
	3- No, never	55.6	61.9	
10- The thought of harming myself has occurred to me	0- Yes, quite often	0.0	0.7	0.387
	1- Sometimes	3.7	2.2	
	2- Hardly ever	4.9	2.2	
	3- Never	91.4	95.0	

**Table 3 jcm-09-02449-t003:** Edinburgh Postnatal Depression Scale (EPDS) results among women hospitalized in the high-risk pregnancy units, during and before the COVID-19 pandemic.

	Hospitalization during the COVID-19 Pandemic *n* = 84 (%)	Hospitalization before the COVID-19 Pandemic *n* = 279 (%)	*p* Value
Total EPDS score ≥ 10	25.0	29.0	0.498
Suicidal ideations (according to question number 10 in EPDS questionnaire)	8.6	5.0	0.221

**Table 4 jcm-09-02449-t004:** Multivariable logistic regression model for the association between timing of hospitalization and EDPS score ≥ 10.

	**Model 1** **EDPS Score > 10**		
	**Adjusted (Odds Ratio (OR))**	**95% {Confidence Interval (CI)}**	***p* Value**
Hospitalization during the COVID-19 pandemic (vs. hospitalization before the COVID-19 pandemic)	1.0	0.52–1.93	0.985
Maternal age (years)	1.0	0.94–1.05	0.916
Ethnicity (Jewish vs. Bedouin)	0.3	0.21–0.66	0.001
Mood disorder	6.1	3.30–11.59	<0.001
	**Model 2** **Suicidal Ideation**		
	**Adjusted OR**	**95% CI**	***p* Value**
Hospitalization during the COVID-19 pandemic (vs. hospitalization before the COVID-19 pandemic)	1.8	0.71–4.85	0.203
Maternal age (years)	0.8	0.81–0.97	0.009

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
