# Peer review of "Risk for Depressive Symptoms among Hospitalized Women in High-Risk Pregnancy Units during the COVID-19 Pandemic"

_jcm, 2020, doi:10.3390/jcm9082449_

Round 1
Reviewer 1 Report
Thank you for the opportunity to review this work. The study is interesting, however I do have multiple concerns that should be addressed by the authors.
In the Methods, please clarify the reason for choosing as comparator period the time between November 2016 and April 2017. Is there a rationale for this, or was the period selected randomly? Since the exposure period was approximately 2 months in the summer period, I wonder why a similar selection in terms of length and seasonality was done for the controls.
The methodology needs extensive revision. Please describe in details the conditions defining a pregnancy as “risky” and in what circumstances women are hospitalized in Israel.
The study needs clarification about the sampling strategy adopted. Were all women asked to participate in the exposed group? And what about the controls? Please expand on the sampling strategy used in the study.
It is unclear to me whether the exposed group completed the EPDS questionnaire at the hospital or retrospectively once they left the hospital; it is also critical to indicate when in pregnancy these women completed the EPDS, as depressive symptoms score may vary during gestation. This is important to keep in mind when comparing exposed and controls.
Please clarify when / how the EPDS was administered to the historical control group. Was another study administering the EPDS conducted back in 2016-2017? This section of the article has to be comprehensive and detailed, so that any other researcher could reproduce the study in another population/setting.
Please describe in more details how data on maternal sociodemographics was collected; does it stem from medical records, or was it self-reported in the questionnaire?
Please indicate whether the study employed the validated version of the EPDS in Israel. Is there any validation study in Israel concerning the cutoff point used in this study (score larger or equal 10?). The cutoff is somewhat low, and 13 is often used. I feel that a cutoff point of 10 may introduce too many false positive, and may be a too weak cutoff. The authors should clearly state whether there is a validated EPDS score cutoff in the Israeli population.
Please indicate to what extent missing data on the items making up the EPDS occurred. Did the authors included only pregnancies whether all 10 items were answered or not? If not, how were missing data handled? Please clarify.
The strategy for model building is to be lacking. On what basis candidate variables were selected for the multivariate model? Was model robustness assessed, and so pre-defined interaction terms? I feel that drawing a casual diagrams to select confounders is more appropriate in this research setting.
Results: what is the response rate in this study?
Table 1: the methods section has given no information about all the variables presented in Table 1 and how these were measured. For instance “mood disorders”: is this variable based on medical record or self-report, is it past or concurrent mood disorder? The variable “chronic illness” should be expanded and possibly analyzed in a more granular way; for instance, is there any data about hypothyroidism?
Table 1, item 10 of the EPDS: is it correct that about 90% of the women had often experienced self-harm thoughts? This seems a very high and unrealistic measure. This mistake becomes evident in Table 3.
In Table 3 another outcome measure is presented about frequency of self-harm; however there is no information in the methods about this measure and how was defined and analyzed.
Table 4: please present crude OR in addition to the adjusted ones. Why the measure on self-harm has not been analyzed? It is unclear how the confounding variables were selected in this study. Please consider drawing a DAG to visualize what factors are possible confounding factors in this setting.
The limitation section has to be expanded; for instance, there is limited reflection about selection bias and external validity of the findings as no response rate is presented. No information about how missing data on the EPDS and confounders were handled; and no reference to possible unmeasured confounders that were not accounted for. For instance, the study does not provide any data about the extent of medication use in these women, including psychotropics.
Reviewer 2 Report
The aim of the study was to assess risk for depression among pregnant women hospitalized during the COVID-19 pandemic compared to women hospitalized before the COVID -19 pandemic. Although the idea is interesting there are some major methodological issues and some minor points that should be considered:
- This is not a cohort study.In particular, this is a cross-sectional study that compared a certain group with a historical cohort of women.
- How was this historical cohort of women found? Is EPDS used as a routine in author’s hospital setting?
- Is EPDS translated and validated for Israel’s population? Why did the authors used 10 as cut-off? This cut-off is usually used for postnatal depression.
- How was the study sample calculated ?
- In general the population and setting section is poorly described.
- According to the results, mood disorders or marital status were not associated with depression.This is weird and is in contrast with the current literature.
- The major part of the introduction (line 42 to 63) is irrelevant with the topic of the manuscript
- Materials: the first paragraph should be moved to the introduction section
Reviewer 3 Report
This study examined the rates of depression in high-risk pregnancy units in Israel both before and during the Covid-19 pandemic. The analyses found no significant difference in depression prevalence between the two groups. The manuscript has a number of strengths, including the clarity of writing, the use of standard depression screening tools, and comparable groups of women across the two time periods. However, I also have some questions and suggestions for improvement, which I detail below:
- Why were rates of depression examined rather than anxiety? As noted in the discussion section, depression often evolves over months, not days or weeks. Leading theories about the emergence of depression emphasize that depression often occurs following repeated stressors or feelings of hopelessness, but the study window is only 7 days and occurs at the onset of the pandemic. Is this enough time for depression to develop?
- I’d like some more information about Covid-19 in Israel. The authors note that as of June 8, 23,813 cases of Covid-19 had occurred in Israel (p. 2). But how many cases were there before the onset of the study? When controlling for population levels, how did the outbreak in Israel compare to other countries at the time of the study? I think this is important information for trying to understand how pressing the issue was in the country during the time when women were being surveyed.
- I think a power analysis would also be informative. Are the sample sizes in each group adequate to seeing expected differences in depression rates?
- The literature review focuses primarily on depression, anxiety, and PTSD increasing after natural disasters. However, as mentioned above, this study is occurring during the onset of a pandemic. Thus, the results might not be comparable. Are there studies about mental health during other health outbreaks (e.g., H1N1 flu, measles, malaria)? The one study that looked at SARS studied the mental health of 19 SARS patients while they were being treated (and was largely qualitative).
- I think it is also worth emphasizing further in the discussion section that the study participants were already hospitalized in high-risk pregnancy units. Therefore, it is possible that their own medical issues would take precedence over the stress of the pandemic, which could account for why there are no group differences. I think it’s important that the results of the current study are not generalized to all pregnant women, as some of them may be experiencing heightened depressive symptoms during this pandemic.
- The authors note that the “vast majority” of people, especially pregnant and postpartum women, experience various form of benefits following terrible events (p. 7). What sort of “benefits” would one expect a pregnant women to experience following a natural disaster or pandemic?
- The clinical implications section conflates resilience with not experiencing depressive symptoms. However, I would caution against defining resilience as not experiencing heightened mental health difficulties after natural disasters or pandemics. Resilience is the ability to overcome difficulties and stressors; it does not mean being completely unaffected by terrible events or not having limited periods of mental health problems. Indeed, the authors note that “resilience” is heavily influence by how closely the disaster personally affected them. By this definition, someone who was evacuated from New Orleans during Hurricane Katrina but did not experience any sustained loss (housing, financial, job, family) and did not develop any mental health problems would be considered “resilient” but someone who experienced depression following the loss of their home, business, and family pet as the result of the flood would be considered “less resilient.” Equating mental health symptoms with resilience in this way contributes to the ongoing stigma around mental health and assigns blame to the individual as some sort of character flaw as opposed to an understandable consequence of experiencing extreme tragedy and loss.
Round 2
Reviewer 1 Report
Thank you for revising the manuscript. I have some additional remarks.
In the abstract and manuscript it is still stated that this is a cohort study, which is incorrect. In this study there is no follow-up of individuals, and so this is not a cohort study. Please change the study type accordingly to a cross-sectional study design.
I feel it is still unclear whether there were missing data on covariates, and how these were handled. If people with missing data are excluded (as in the case for the EPDS), then this is a limitation that should be acknowledged. Depending on the mechanism of missing data, a complete case approach may introduce bias.
Please consider revising the Title from "risk for depression" to "risk for probable depression", or alternatively "risk for depressive symptoms" as the main outcome variable is based on a screen tool.
Reviewer 2 Report
I would like to thank the authors for the detailed answers. However, my major concern is about my previously raised point 2 regarding the methodology of the population found: How was this historical cohort of women found?
The authors replied that "Data regarding the control group was collected in part of a previous prospective study that was performed in the same high-risk pregnancy units (12)".
However, this reference is an old (2008) paper by Brandon et al describing a study conducted in Baylor University Medical Center Antepartum Unit in Dallas, Texas. How is that related to Soroka University Medical Center in Israel?
